# Interventions with Music in PECTus excavatum treatment (IMPECT trial): a study protocol for a randomised controlled trial investigating the clinical effects of perioperative music interventions

Ryan J Billar [ID],[1] A Y Rosalie Kühlmann,[2] J Marco Schnater,[1] John Vlot,[1] Jeremy J P Tomas,[3] Gerda W Zijp,[4] Mandana Rad,[5] Sjoerd A de Beer,[6] Markus F Stevens,[7] Marten J Poley,[1,8] Joost van Rosmalen,[9] Johannes F Jeekel,[10] Rene M H Wijnen[1]

For numbered affiliations see end of article.

**Correspondence to**
Ryan J Billar;
r.billar@erasmusmc.nl

## ABSTRACT

**Introduction** Pectus excavatum repair is associated with substantial postoperative pain, despite the use of epidural analgesia and other analgesic regimens. Perioperative recorded music interventions have been shown to alleviate pain and anxiety in adults, but evidence for children and adolescents is still lacking. This study protocol describes a randomised controlled trial that evaluates the effects of recorded music interventions on postoperative pain relief in children and adolescents after pectus excavatum repair.

**Methods** A multicentre randomised controlled trial was set up comparing the effects of perioperative recorded music interventions in addition to standard care with those of standard care only in patients undergoing a Nuss procedure for pectus excavatum repair. One hundred and seventy subjects (12–18 years of age) will be included in three centres in the Netherlands. Patient inclusion has started in November 2018, and is ongoing. The primary outcome is self-reported perceived pain measured on the visual analogue scale. Secondary outcomes are anxiety level, analgesics consumption, vital parameters such as heart rate, blood pressure and respiratory rate, length of hospital stay, postoperative complications, quality of life and cost-effectiveness.

**Ethics and dissemination** This study is being conducted in accordance with the Declaration of Helsinki. The Medical Ethics Review Board of Erasmus University Medical Centre Rotterdam, The Netherlands, has approved this protocol. Results will be disseminated via peer-reviewed scientific journals and conference presentations.

**Trial registration number** NL6863

## INTRODUCTION

Pectus excavatum (PE) is the most common congenital chest wall deformity affecting 0.1%–0.8% of live births, affecting boys more than girls. Operative repair is indicated when symptoms or signs of heart and/or lung dysfunction are present,[1] or when the patient is much concerned about the cosmetic appearance and psychosocial problems occur.[2 3] The optimal age for repair is between 12 and 16 years.[4] Numerous surgical techniques have been developed to correct PE, of which the Nuss procedure is now among the most commonly employed techniques.[5 6] The Nuss procedure involves inserting a convex steel bar beneath the sternum to reposition the sternum anteriorly and thereby effectively correcting the deformity.[7] It is associated, however, with substantial postoperative pain, despite the use of epidural analgesia or patient-controlled intravenous opioid administration.[8 9] Pain management is the critical component of postoperative care

as postoperative pain has implications for activity and quality of life[10] and is the primary factor determining the length of hospital stay.[11]

### Strengths and limitations of this study

► This study is the first multicentre randomised controlled trial evaluating the effects of recorded music interventions on pain experience in older children before, during and after pectus excavatum repair with the Nuss procedure.

► Data will be collected during hospitalisation and up until 3 months postoperatively to shed light on the effect of perioperative recorded music interventions during hospitalisation and after hospitalisation evaluating both short-term and potentially long-term effects.

► The study participants and participating surgeons are not blinded to the interventions, which is a limitation; however, the anaesthesiologists and pain specialists will be blinded to the study arm allocation.

Therefore, interest is growing in finding new ways to alleviate postoperative pain, such as perioperative music interventions. In previous studies in adult surgical patients, recorded music interventions reduce pain medication consumption and improve the management of pain and anxiety.[12–20] However, in children and adolescents undergoing surgery, a definite conclusion about the effect of recorded music interventions has yet to be drawn.[21] Especially in paediatric surgical procedures associated with substantial postoperative pain, such as the Nuss procedure, music interventions might be effective in reducing children's pain and anxiety. We designed a multicentre randomised controlled trial (Interventions with Music in PECTus excavatum treatment (IMPECT) trial) to evaluate whether adjuvant recorded music interventions are indeed associated with less postoperative pain in children and adolescents undergoing the Nuss procedure for PE repair.

## METHODS AND ANALYSIS
### Study design
The IMPECT trial is a randomised controlled trial with two study arms, designed to compare the effects on postoperative pain of perioperative recorded music interventions in addition to standard care (intervention group) versus standard care (control group)—prior, during and after the Nuss procedure for PE repair. We will include 170 subjects of children and adolescents (12–18 years of age) operated on in three centres in the Netherlands: the Erasmus University Medical Centre-Sophia Children's Hospital, Rotterdam; Haga Hospital-Juliana Children's Hospital, The Hague; and Academic Medical Centre-Emma Children's Hospital, Amsterdam. We started enrolment in November 2018. The first patient included was in January 2019. This study protocol follows the Standard Protocol Items: Recommendations for Interventional Trials (SPIRIT) guidelines (see SPIRIT checklist in online supplementary material). The underlying protocol follows the Consolidated Standards of Reporting Trials (CONSORT) guidelines for non-pharmacological treatments. This trial was registered on trialregister.nl.

### Randomisation, blinding and treatment allocation
A parallel randomisation with equal allocation ratio is being carried out to individually allocate subjects to either the intervention or the control group. An online web-based randomisation program (ALEA; FormVision, Abcoude, The Netherlands) generates the random allocation sequence by the use of random block size randomisation and is stratified by centre with an equal allocation ratio per centre in both study arms. Allocation concealment will be ensured, as the service will not release the randomisation code until the patient has been recruited into the trial. The anaesthesiologists and pain specialists involved do not have access to the randomisation program and are blinded to the subject's study arm allocation, as well as the person analysing the data.

### Interventions
Subjects in the intervention arm receive a recorded music intervention prior to and during surgery and postoperatively the first 3 days (see figure 1). The music intervention prior to surgery is 30 min long and takes place before the administration of premedication. After induction of general anaesthesia and after final positioning of the patient a headphone with music is applied and will remain during surgery. The headphones will be removed at the recovery unit, when patients are fully awake. The music interventions after surgery are each 30 min long and take place twice a day, in the morning and evening. In each hospital, the best times to start the intervention will be established to assure blinding of both the anaesthesiologists and pain specialists. Subjects in the control arm rest for 30 min prior to surgery and the administration of premedication, and wear a headphone without music during surgery. After surgery, they receive regular postoperative care without music interventions. Subjects in the control group are instructed to refrain from much listening to music during the hospital stay. The subjects in both control group and intervention group are requested to self-document all activities performed, such as listening to music, playing video games, using the computer and watching movies and television. Participation ends at the scheduled postoperative check-up at the outpatient clinic (see figure 1). All study measurements take place during hospitalisation and at the outpatient clinic. No extra visits to the hospital are required.

### Music selection
It has been suggested that individual music preference is important to the effect of a music intervention.[22] Nevertheless, a study has shown that playing music from a preselected playlist by the researcher has the largest beneficial effect on postoperative pain, compared with the subject's own favourite music or preselected music without taking the music preference of the subject into account.[19] However, definite conclusions in this regard cannot be drawn. Furthermore, research in rodents suggests that loud rock music may have a negative effect and may act as a stressor.[23]

Therefore, in collaboration with a specialised music therapist we have composed three music playlists without loud rock music, which the subject can choose from. The playlists are categorised into three different genres of music: pop, lounge and classical music. Subjects can choose from either of these playlists. Subjects may choose a different playlist during surgery. Music will be heard through an on-ear headphone connected to a digital music player. After surgery, subjects in the intervention group may listen to their own preferred music. Approval from Buma/Stemra, the Dutch collecting society for

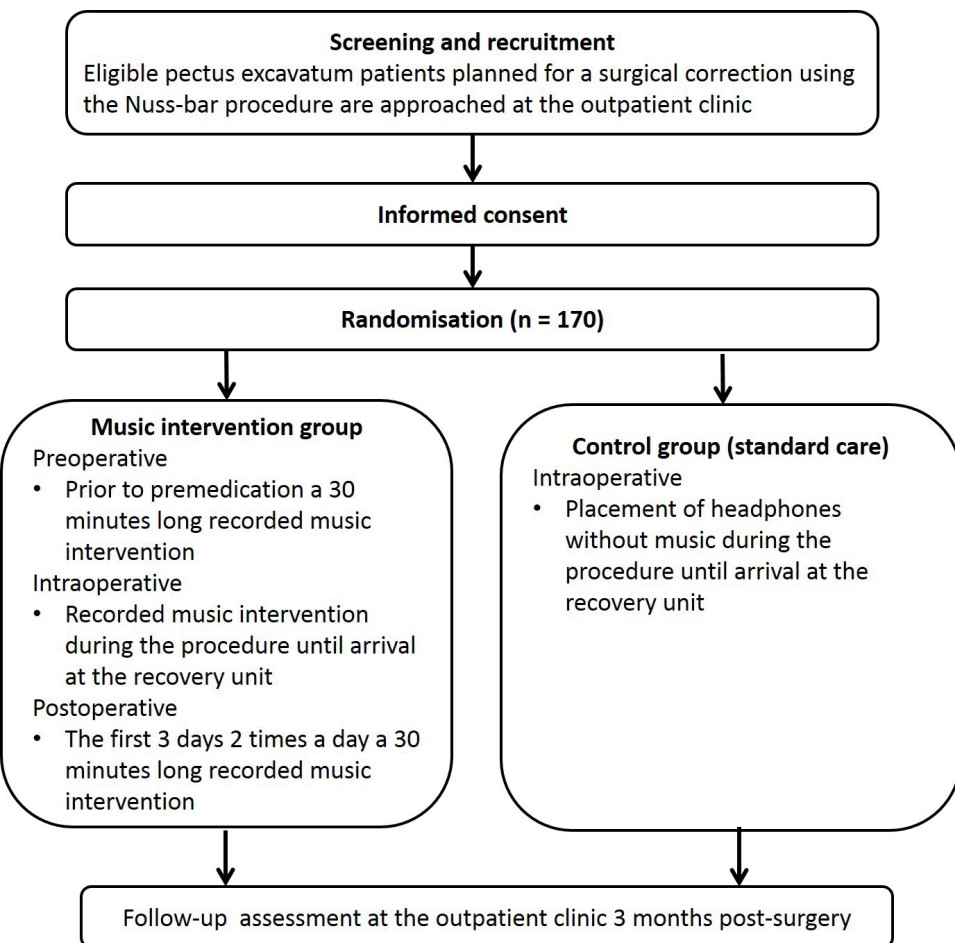

**Figure 1** Flow chart of study interventions and assessments.

composers and music publishers, has been received to use any licensed music.

### Anaesthetic treatment

There is no nationwide standard anaesthesia protocol for the Nuss procedure in the Netherlands. Therefore anaesthesia protocols differ between centres. Randomisation should control for such variation between centres. However, it will be analysed statistically.

All centres apply EMLA cream at the intravenous line insertion site. Furthermore, all patients receive epidural analgesia, which are preferably placed at fifth thoracic level. All centres used long-acting local anaesthetics with an adjuvant epidurally. General anaesthesia was induced and maintained by propofol combined with opioids and neuromuscular relaxation induced by rocuronium. Postoperative analgesia was maintained with a continuous epidural infusion of a long-acting local anaesthetic (ropivacaine 0.2% or bupivacaine 0.125%) with an adjuvant. All patients received weight-based doses of paracetamol and a non-steroidal anti-inflammatory drug postoperatively. After epidural removal pain was treated by oral opioids as required.

However, there are some major differences between hospitals: the Emma Children's Hospital gives standard premedication with clonidine 150 µg and 300 mg gabapentin, while the other two hospitals do not give any pharmacological premedication. Furthermore, in Emma Children's Hospital patients receive gabapentin 300 mg twice daily for 5 days and receive patient-controlled analgesia with morphine in addition to the epidural catheters. Finally, while Juliana and Sophia Children's Hospitals use sufentanil 0.5 µg/mL as an epidural adjuvant, Emma Children's Hospital uses clonidine 1 µg/mL.

### Outcome parameters

The primary outcome parameter is pain, defined as the average pain score, as measured by a visual analogue scale (VAS-pain), that patients will report at the third day postoperatively. The scale of the VAS-pain varies from 0 to 100, whereas 0 is defined as no pain and 100 as the worst pain imaginable. This scale has been recommended and validated for the measurement of acute pain in children 8 years of age and above and is also sensitive to changes in pain levels postoperatively.[24–26]

Secondary outcome parameters include:

► The morphine consumption in the first 3 days postoperative as calculated by the morphine equivalent daily dose/kilogram (MEDD/kg) and the consumption of other analgesics in milligrams.

- Physiological variables such as heart rate, blood pressure and respiratory rate will be measured throughout their hospital stay.
- Levels of anxiety and distress will be measured before surgery through the State-Trait Anxiety Inventory (STAI) for children. This questionnaire consists of two separate 20-item self-report rating scales for measuring trait and state anxiety. The trait anxiety is a relatively stable personality disposition, while state anxiety is the situation-related anxiety and this may differ depending on the stress of a particular moment.[27] The questionnaire has been translated into Dutch and has been validated.[28]
- Quality of life will be measured before surgery and at their first check-up at the outpatient clinic through the Child Health Utility Questionnaire (CHU9D). This validated questionnaire consists of nine items that assess the child's functioning across domains of worry, sadness, pain, tiredness, annoyance, schoolwork/homework, sleep, problems with daily routine and ability to join in activities.[29–31]
- Postoperative complications and length of hospital stay are recorded.
- The subject's postprocedural pain after 3 months will be evaluated with the VAS, the CHU9D and the 'TNO questionnaire for sport and physical activity'. This validated Dutch questionnaire assessed a person's daily activities.[32] This questionnaire serves to measure rehabilitation as a derivative of the postprocedural pain. Baseline measurements for the 'TNO questionnaire for sport and physical activity' will also be performed before surgery.
- Considering the potential influence of pain and the use of analgesics on length of stay, cost-effectiveness of the intervention will be determined through a cost-utility analysis.

### Eligibility criteria

Potential subjects visiting the outpatient clinic of the three paediatric surgery departments involved will be informed about our study. A member of the research team undertakes the initial screening for eligibility. The following inclusion criteria apply:

- Ages 12–18 years.
- Scheduled for primary PE repair according to the Nuss procedure with either one or multiple bars.
- Postoperatively, initial placement of a thoracic epidural or both thoracic epidural and patient-controlled analgesia system.
- Good knowledge of the Dutch language, by both patients and parents.
- Written informed consent. Additional written informed consent by parents or legal guardian is only necessary for children under the age of 16 years.

  The following exclusion criteria apply:

- Hearing impairment.
- Secondary PE surgery or other prior thoracic surgery.
- Known severe mental or psychiatric disorder.

- Known impaired communication with patient and parents as collected.
- Presence of chronic pain syndrome: ongoing pain lasting longer than 3 months or ongoing pain lasting longer than the reasonably expected healing time for the involved tissues.

One week after being informed about the study, eligible subjects will be called by telephone to inquire if they wish to participate.

### Sample size

A power calculation was performed by the Department of Biostatistics of the Erasmus Medical Centre for the primary outcome parameter: pain, defined as the average pain score, as measured by a VAS, that patients will report at the third day postoperatively. Evidence on the effects of recorded music interventions prior, during and after surgery in PE repair is lacking. However, a recent meta-analysis, which investigated music interventions on pain in surgical patients, found an overall effect size measured as the Cohen's delta of −0.50 (95% CI −0.66 to −0.34).[19]

We assumed a low correlation between the VAS score preoperatively and postoperatively of 0.3. Thus, to obtain a power of 90% using a two-sided significance level of p<0.05, each study arm requires 77 subjects. To account for dropouts, we will include 85 subjects per study arm, resulting in a total sample size of 170.

### Statistical analysis

The main study endpoint will be the VAS-pain score reported by the subject during the length of hospital stay, three times a day. The mean VAS-pain score of each day will be calculated per subject. The mean VAS-pain scores between the music and control group on the third day will be compared with an analysis of covariance (ANCOVA) test, with adjustment for the effects of centre and baseline VAS-pain score. The main analysis will be based on the intention-to-treat principle. In case of non-compliance, a sensitivity analysis will be performed using per-protocol analyses. A two-sided p value <0.05 will be considered to be statistically significant. For the primary outcome parameter, only the available data will be analysed (no imputation of missing data).

In a sensitivity analysis, we will also adjust for possible confounder variables in the linear regression model for the following variables: age, gender, body mass index and epidural use. Finally, we will also perform a second sensitivity analysis to determine if the effectiveness of the intervention depends on the type of music chosen, by adding these genres as categories to the linear regression model.

The VAS score of each time point will be analysed using a linear mixed model, with the baseline value (observed before surgery), group (control arm or intervention arm), centre and time point, and the interaction between group and time point as independent variables. Total consumption of analgesics and type of analgesics in milligrams will be added to the analyses. Also an interaction effect of centre and group will be examined due to variation in

anaesthesia protocols in the participating centres. Using information criteria, it will be determined if it is necessary to add a random intercept and/or random slope of time point to this model, to account for the within-subject correlations. If required, a transformation of the outcome will be applied to ensure normality of the model residuals.

The secondary outcome parameters will be analysed as follows:

► MEDD/kg and total dosage of other analgesia.

There may be differences between centres in usage of patient-controlled analgesia and epidural anaesthesia. Therefore, the difference between the intervention group and the control group will be tested using multiple linear regression, with adjustment for the effects of centre. When necessary, an appropriate transformation of the outcome (MEDD/kg) or total dosage of other analgesia in milligrams will be performed to achieve a normal distribution of the residuals.

► Score on STAI questionnaire and health-related quality of life (HRQoL).

The scores of the STAI and HRQoL questionnaires will be compared between groups using ANCOVA, with group, centre and the baseline STAI score and HRQoL score before the intervention or the resting period as independent variables.

► Physiological measurements, including blood pressure, heart rate and respiratory rate.

These variables will be analysed using a linear mixed model, with the baseline value (observed before surgery), group (control arm or intervention arm), centre, time point, and the interaction between group and time point as independent variables.

Using information criteria, it will be determined if it is necessary to add to a random intercept and/or random slope of time point to this model, to account for the within-subject correlations. If necessary, a transformation of the outcome will be applied to ensure normality of the model residuals.

► Complications, like postoperative ileus (number of days), nausea/vomiting (number of days and also antiemetics used) and pruritus.

The duration of postoperative ileus, nausea and vomiting will be compared between groups using a Mann-Whitney test, stratified by centre (ie, a Van Elteren test). The percentage of patients with pruritus will be compared between groups using a stratified $\chi^2$ test.

► Length of hospital stay (number of days).

The length of hospital stay will be compared between groups using a Mann-Whitney test, stratified by centre (ie, a Van Elteren test).

## Economic evaluation

We will analyse the cost-effectiveness of the music intervention versus 'standard care' from a healthcare perspective, using the techniques of cost-effectiveness analysis and cost-utility analysis and following recommended methods for economic evaluations.[33]

Medical costs (ie, costs within the healthcare sector) will be analysed, including costs of surgeries, hospital days (on the ward or intensive care unit), medications (such as analgesics), diagnostic radiography and intercollegiate consultations. For the intervention group, costs of the music intervention will be added, mainly consisting of a Spotify subscription. In addition, costs of healthcare use after the initial hospitalisation will be calculated (eg, outpatient visits, consultations by telephone, (pain) medication and rehabilitation). Resource consumption for all these items will be derived from electronic databases at the participating centres and from a questionnaire (based on the iMTA Medical Consumption Questionnaire).[34] Unit prices (calculated using economic cost prices or standard prices) will be multiplied by the quantities for each resource used, and then summed over the separate types of resource to give a total cost per patient. Non-medical costs (eg, out-of-pocket costs and costs of productivity losses incurred by the parents) will be ignored in this study, as these are expected to be relatively minor and not to differ notably between the study groups.

Regarding the patient outcomes, the economic evaluation will look at pain (as measured by the VAS) and HRQoL measured by the CHU9D. The CHU9D is a preference-based measure of HRQoL allowing for the calculation of quality-adjusted life-years (QALY), which is a commonly used health outcome measure to calculate the benefits of new interventions within cost-utility analyses for economic evaluation. QALYs will be calculated based on the CHU9D and using linear interpolation between measurement points.

Building on these data on costs and patient outcomes, incremental cost-effectiveness ratios will be calculated, expressed as incremental costs to obtain a reduction of one additional unit (10 mm or 1 cm) in the VAS score and as incremental costs per QALY gained. Otherwise, the economic evaluation will focus on dominance of one treatment over the other with respect to lower cost and greater effect. The time horizon of the analysis will be the 3 months' follow-up period (starting at the beginning of the hospital admission for the PE repair). As a consequence, discounting will not be necessary. Analysis of uncertainty is illustrated through cost-effectiveness planes (via bootstrapping). Sensitivity analysis will be performed to assess the robustness of the analysis to certain assumptions.

## Trial monitoring

An independent trial monitor overseeing all aspects of design, delivery and quality assurance has been appointed by the sponsor, the head of the Department of Paediatric Surgery of the Erasmus Medical Centre. The trial will be monitored at least once per year and a written monitor report will be submitted to the sponsor after each trial site visit or trial-related communication.

## Data management

Participant data are stored on a secure database in accordance with the General Data Protection Regulations (2018). Data are handled confidentially, deidentified and coded with a unique study number. Published data from this study cannot be traced to a specific subject. Data management for the study was done through OpenClinica and LimeSurvey. Study staff assigned to manage data has access to the OpenClinica and LimeSurvey application and is required to log in via an individualised username and password combination. Study staff located at other institutions only has access to the data collected at their sites. The local investigators will safeguard the key that links the unique study number to the patient's name at a separate server.

Trial documentation and data will be archived for at least 10 years after completion of the trial.

## Patient and public involvement

Patients undergoing PE repair prior to the start of this study evaluated and helped us in composing our preselected music playlists.

## ETHICS AND DISSEMINATION
### Ethics

This study protocol has been reviewed and approved by the Medical Ethics Review Board of the Erasmus Medical Centre in Rotterdam on 5 September 2018. This study is being conducted according to the principles of the Declaration of Helsinki (64th WMA General Assembly, Fortaleza, Brazil, October 2013) and in accordance with the Medical Research Involving Human Subject Act (Dutch: WMO). The trial is registered with the Netherlands Trial Register. To prohibit playing music on the operating room and testing the epidural sensory block daily were approved and implemented as a minor amendment on 9 October 2019.

### Benefits and risks assessment, group relatedness

There are no risks associated with listening to music, except potential hearing damage. To prevent hearing damage, the music administered on the headphones will be adjusted to a maximum of 60 dB, which is the advised loudness of a music intervention in medical care.[35] The maximum decibels advised to be exposed to for 40 hours/week is 80 dB.[36] Therefore, risk of participation can be considered negligible and the burden minimal. During the informed consent process, it will be made clear that participation in this study has no direct benefits to the patient, and that refusal to participate will not have impact on the care received by any of the medical staff. PE is preferably corrected at ages 12–18. This study therefore cannot be conducted without the participation of this group.

All adverse events will be documented. Music intervention itself, however, is considered harmless and safe. Therefore, we expect no intervention-related serious adverse events or any other disadvantages for participants in this study.

## Dissemination

The research team is committed to full disclosure of the results of the trial. Findings will be reported in accordance with CONSORT guidelines and we aim to publish in high-impact journals. Given the multitude of outcome parameters, results will be divided over several papers. The funder will take no role in the analysis or interpretation of trial results.

**Author affiliations**
[1]Pediatric Surgery, Erasmus MC Sophia Children's Hospital, Rotterdam, Zuid-Holland, The Netherlands
[2]Anaesthesiology, Saint Antonius Hospital, Nieuwegein, Utrecht, The Netherlands
[3]Anaesthesiology, Erasmus MC Sophia Children's Hospital, Rotterdam, Zuid-Holland, The Netherlands
[4]Paediatric Surgery, Haga Hospital Juliana Children's Hospital, Den Haag, Zuid-Holland, The Netherlands
[5]Anaesthesiology, Haga Hospital Juliana Children's Hospital, Den Haag, Zuid-Holland, The Netherlands
[6]Paediatric Surgery, Emma Children's Hospital AMC, Amsterdam, North Holland, The Netherlands
[7]Anaesthesiology, Emma Children's Hospital AMC, Amsterdam, North Holland, The Netherlands
[8]Institute for Medical Technology Assessment, Erasmus University Rotterdam, Rotterdam, The Netherlands
[9]Biostatistics, Erasmus MC, Rotterdam, Zuid-Holland, The Netherlands
[10]Neuroscience, Erasmus MC, Rotterdam, The Netherlands

**Contributors** Each author has contributed significantly to, and is willing to take public responsibility for, one or more aspects of the study. JFJ and RMHW conceived the study idea. RJB coordinated the research protocol and wrote the first draft of the manuscript. RJB, AYRK, JMS, JV, JJPT, GWZ, MR, SAdB, MFS, MJP, JvR, JFJ and RMHW critically revised the manuscript. All authors have seen and approved the final version of the manuscript being submitted. The article is the authors' original work, has not received prior publication and is not under consideration for publication elsewhere.

**Funding** This research is funded by the Erasmus Medical Centre, Rotterdam, The Netherlands.

**Competing interests** None declared.

**Patient and public involvement** Patients and/or the public were involved in the design, or conduct, or reporting, or dissemination plans of this research. Refer to the Methods section for further details.

**Patient consent for publication** Not required.

**Ethics approval** The study protocol has received ethical approval from the Medical Ethical Review Committee of the Erasmus Medical Centre in Rotterdam prior to the beginning of the study.

**Provenance and peer review** Not commissioned; externally peer reviewed.

**ORCID iD**
Ryan J Billar http://orcid.org/0000-0002-3624-8222

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
