## [Reviewer comments · BMJ Open]

ARTICLE DETAILS

TITLE (PROVISIONAL)	Interventions with Music in Pectus Excavatum Treatment (IMPECT trial): A study protocol for a randomized controlled trial investigating the clinical effects of perioperative music interventions
AUTHORS	Billar, Ryan; Kühmann, Rosalie; Schnater, Marco; Vlot, John; Tomas, Jeremy; Zijp, Gerda; Rad, Mandana; de Beer, Sjoerd; Stevens, Markus; Poley, Marten; van Rosmalen, Joost; Jeekel, Johannes; Wijnen, Rene

VERSION 1 – REVIEW

REVIEWER	Gary Elkins Department of Psychology and Neuroscience Baylor University Waco, TX USA
REVIEW RETURNED	22-Dec-2019

GENERAL COMMENTS	This is a well-written study protocol for a multi-site randomized clinical trial of music intervention and standard care vs standard treatment without music. Accrual for the study began in November 2018 and is on-going. The protocol states that an independent trial monitor has been appointed to oversee all aspects of the design, delivery, quality assurance and data analysis. However, there is no information provided on who appointed the monitor and who the monitor reports to regarding quality assurance and protocol adherence. The authors state that "there are no risks associated with music, except hearing damage" and that the music will be delivered via headphones adjusted to a maximum of 60 dB. It is unclear if adverse events will be inquired about and documented. While serious adverse events caused by the intervention are very unlikely, the study provides an opportunity to determine any adverse events related to the study involvement. The study flowchart is clear and informative. The SPIRIT checklist is completed and accurate.
---

REVIEWER	Iain Moppett University of Nottingham UK
REVIEW RETURNED	27-Jan-2020

GENERAL COMMENTS	The authors have submitted a trial protocol for an ongoing RCT of music therapy. The trial is ongoing so my comments are mainly questions for the authors rather than suggestions for significant changes.  1) There is a very minor discrepancy between the trial registration and the paper (Dutch vs Dutch or English). 2) The study appear to be nearing completion (the end date on the registry is 2020-06-01 (June I assume). How many participants have been recruited? 3) Is this a trial of the addition of music to 'normal' care, or is it really a trial of removal of music? It would be helpful to have some context for how many of this population listen to music before and after surgery outside of the trial. In- and on-ear headphones are ubiquitous on the wards. 4) How is pain being assessed - dynamic or static? 5) Is STAI valid in teenagers? 6) HRQOL seems to be used as a synonym for CHU9D - is this appropriate? 7) I can't replicate the power analysis -- my estimate is 85 per group before drop outs (using two different calculators). 8) I'm slightly unclear as to whether the primary outcome is the mean VAS on day three, or the mean over the three days. 9) Is postoperative ileus a common / relevant complication for this procedure? 10) I am not an expert in health economic evaluation, so suggest this is reviewed by such a person. At the least there needs to be some supporting evidence for the use of CHU9D for calculation of QALYs in the Dutch population. 11) A minor point, but there are varying uses of tense to describe what is happening / has happened / will happen. Thank you for asking me to review this interesting project. As always, these are simply my opinions, the authors may well disagree.
--

VERSION 1 – AUTHOR RESPONSE

Reviewer 1, Gary Elkins:

- An independent trial has been appointed by the sponsor, the head of the department of paediatric surgery of the Erasmus Medical Centre. The trial will be monitored at least once per year and a written monitor report will be submitted to the sponsor after each trial-site visit or trial-related communication. This information has been added to the manuscript.
- We will document every adverse event. We have added this information to our manuscript.

Reviewer 2, Iain Moppett:

1. Discrepancy is resolved. We have updated the trial registration, as it was not up to date.
2. 32 participants have been recruited up until this moment. At the moment we have also filed an extension of the inclusion period until December 2022. We expect approval from our Medical Ethics Review Board this month.
3. This trial is about the addition of music, while participants in the control group listen to music as little as possible. In our case report forms we document how much each participant listen to music

before participation of our trial, during hospitalization and after hospitalization. Participants in our music group have fully access to our on-ear headphones if they don't have any themselves.

4. Pain is measured three times a day statically as a Visual Analogue scale.

5. Different versions of the STAI exist. The Dutch translation of the STAI in children is validated until the age of 15 years. As the average age is between 15 and 16 years and to make the results as comparable as possible, we have chosen to use the STAI for children only.

6. The CHU9D is a questionnaire that measures the Health Related Quality of Life.

7. Using an independent samples t-test, the required sample size to compare the postoperative VAS between groups with a Cohen's d of 0.50 would indeed be equal to 85 or 86 patients per group. Our primary analysis is however not based on a t-test, but on an ANCOVA test, which achieves more power and lower sample size than a t-test by taking into account the correlation between the preoperative and the postoperative VAS. With a correlation of 0.30, the required sample size is 77 per group for the ANCOVA test.

To clarify the type of statistical test in the description of the power analysis, we have inserted 'with an ANCOVA test' after the words 'Thus, to obtain a power of 90%'

8. The primary outcome is the mean VAS on day 3. We have reformulated the text in the manuscript to avoid confusion.

9. Due to the very high use of opioids postoperatively in this patient population we have identified postoperative ileus as a relevant complication.

10. A health economic expert is assigned to our study to assure the validity of our health economic evaluation.

11. We have reviewed the article again to assure a more consistent use of the varying tenses.

Formatting amendments:

- In the Netherlands it is common that the Christian baptised names are part of the initials. These Christian baptised names however, are different from the first names. This explains the discrepancy between the initials and the first names between our manuscript and scholar one.

VERSION 2 – REVIEW

REVIEWER	Iain Moppett University of Nottingham, UK
REVIEW RETURNED	07-May-2020

GENERAL COMMENTS	The authors have made sensible changes. I disagree with them that this is a trial of music, rather than a trial of denial of music, but It's not my study.
--